# Determining the Mechanical Properties of Solid Plates Obtained from the Recycling of Cable Waste

**DOI:** 10.3390/ma15249019

**Published:** 2022-12-16

**Authors:** Maciej Wędrychowicz, Władysław Papacz, Janusz Walkowiak, Adam Bydałek, Andrzej Piotrowicz, Tomasz Skrzekut, Jagoda Kurowiak, Piotr Noga, Mirosław Kostrzewa

**Affiliations:** 1Institute of Materials and Biomedical Engineering, Faculty of Mechanical Engineering, University of Zielona Gora, Prof. Z. Szafrana Street 4, 65-516 Zielona Góra, Poland; 2Institute of Mechanical Engineering, Faculty of Mechanical Engineering, University of Zielona Gora, Prof. Z. Szafrana Street 4, 65-516 Zielona Góra, Poland; 3Faculty of Non-Ferrous Metals, AGH University of Science and Technology, A. Mickiewicza Av. 30, 30-059 Cracow, Poland; 4Eko Harpoon Recycling sp. z o. o., Cząstków Mazowiecki 128, 05-152 Czosnów, Poland

**Keywords:** plastic materials, recycling/recovery, mechanical processing, mechanical properties

## Abstract

In this article, the possibility of obtaining a solid plate from waste cable sheaths, by mechanical recycling, i.e., grinding, plasticising and pressing, is discussed—waste cable sheaths being pure PVC with a slight admixture of silicone. Press moulding was carried out under the following conditions: temperature 135 °C, heating duration 1 h and applied pressure 10 MPa. The yield point of the obtained solid plate obtained was 15.0 + −0.6 MPa, flexural strength 0.94 MPa, yield point 0.47 MPa and Charpy’s impact strength 5.1 kJ/m^2.^ The resulting solid plate does not differ significantly from the input material, in terms of mechanical strength, so, from the point of view of strength, that is, from a technical point of view, such promising processing of waste cables can be carried out successfully in industrial practice.

## 1. Introduction

Due to the resistant degradation process and thus negative impact on nature and also the huge post-consumption volume, plastic material wastes are a severe environmental problem. At present, the emission of plastics into the environment is intense and continuous. We can distinguish the “anthroposphere”, i.e., a separate synthetic-origin geological stratum rich in components that do not occur naturally in nature, which includes the so-called microplastics; it is detected not only in the anthroposphere but also in the biosphere and penetrates into water reservoirs, the soil and the air [1,2].

Plastic waste consists of polymers such as ethylene terephthalate (PET), high-density polyethylene (HDPE), low-density polyethylene (LDPE), polyvinyl chloride (PVC), polyamides (PA) and polystyrene (PS) [3,4], etc. Polymers are large-molecule synthetic materials made up of long chains of repeating units called monomers, consisting of carbon, hydrogen and heteroatoms such as chlorine, nitrogen and others. Due to the possibility of large-scale production, polymers are produced primarily from petroleum-based substances but less often from secondary materials. The continuous demand for plastics has made the accumulation of plastic waste in landfills take up a lot of usable space; in addition, the growing demand for plastics has led to the depletion of crude oil as part of a non-renewable fossil fuel. The simplified structure of polymers, especially in electronic devices, and the low cost of plastic-including products results in a short period of use (product life cycle), after which they become waste. The production of solid plastic waste is increasing day by day and is now around 150 million tonnes per year [5,6]. The recycling of plastics plays an important role in the management of polymer waste, thus allowing the recovery of secondary raw materials in the form of regranulates or chemicals. A crucial step in achieving a circular economy is recycling with high added value [7,8]. One of the most critical steps in waste management, in terms of cost and as material recovery efficiency, is the segregation and separation of waste. 

### 1.1. Comparative Analysis of Mechanical Properties of Recycled Plastics and Economic Justification for Their Use 

An analysis [9] of the profitability, the profit and loss of the investment in a technological line for the mechanical recycling of PVC from cable waste, was carried out over a 3-year period from the start of the investment. The assumptions of this analysis were as follows: annual PVC processing in a range of 803–972 tons, the amount of waste generated 20–24 tons, the cost of investment 25 thousand USD and operating expenses 5 thousand USD. Already after the second year, the investment is becoming profitable with a profit of 24 thousand USD, which almost completely compensates for the capital expenditure of the first year. Several other factors, such as the annual increase in the price of PVC, the costs associated with the disposal of PVC and the investment leverage, make the investment even more profitable with the profit in the third year being estimated at over 26 thousand USD. Of course, such an analysis should be conducted individually for each recycling plant, as this takes into account such aspects as local laws and taxes which differ depending on the region. Nevertheless, it shows clearly that the idea of mechanically recycling PVC from cable waste can be economically justified.

Although recycled materials may have physical properties similar to primary plastics, the resulting financial savings are negligible because the mechanical properties of most plastics deteriorate significantly after many processing cycles [10,11], which may contradict the above paragraph somewhat and constitute a counterpoint to it. One characteristic of the so-called mechanical recycling of plastics is that the material is degraded along with the next processing cycle [12]. Moreover, the permanent cross-linking of polymer structures is not suitable for their multiple processing, but these can be used, for example, as an additive to modified asphalt materials, contributing to the overall improvement in the material [13,14,15,16]. In order to obtain higher mechanical properties of the material, appropriate modification should be applied, e.g., physical modification [17,18,19,20,21]. The studies published so far have shown that the application of an appropriate compatibility method can be successfully used to modify a regenerated rubber mixture, containing various thermoplastics [22]. On the other hand, utility items, made partially or entirely from recycled materials can be used wherever excessive and specific physico-chemical properties are not required or where a short lifetime is expected for such items. In such a case, the use of a secondary material for the production of products can be much more commercially and economically justified; however, the increasing reduction in the value of the material, in such a production cycle scenario, should be taken into consideration. In the research published so far, it is necessary to mention the references that have shown the high efficiency of combining waste polyethylene with a copolymer and an elastomer in manufacturing a material that can be effectively used in industry [23,24]. Marossy [25] and Záraga [26] analysed the effect of chlorinated polyethylene in PVC/PE blends and found that polyethylene improves its rigidity if the concentration of chlorinated polyethylene is not too high. Kollar and co-authors [27] demonstrated that a blend of polyvinyl chloride with polyethylene can be obtained by melt blending using high- and low-density polyethylene, without the use of a stabilizer. Xu [28] studied the separation of the plasticiser of polyvinyl chloride from copper wires obtained from thin electrical cables and investigated the effect of the plasticiser before and after extraction, finding similar properties. 

### 1.2. Recycling of Cable Waste and Its Use 

A large part of the research on copper cable recycling is focussed on developing better technological and organisational solutions so as to be able to recover as many material fractions as possible. The problem is that the recovered waste is not suitable for industrial use. Therefore, numerous studies are carried out on the development of a method to improve the properties of polymer recyclates, in order to find suitable industrial applications for them. Currently, plastic recyclates are most often used in the process of injection and extrusion, which do not yet have their equivalent in industrial usage [29,30]. Recycling cables from a scrap of electric and electronic equipment is a complex problem because there is a wide range of plastics from standard, pure thermoplastics to cross-linked polymers with their numerous mixtures such as glass and carbon fibre, talc or calcium carbonate. In order to reduce the costs related to segregation, it can be proposed to eliminate the costly separation in favour of processing mixed waste into full-value products using hot pressing technology [31,32]. Several works [33,34,35] deal strictly with research on the mechanical recycling process and the strength properties of the utility materials obtained from PVC with a varied history of use and forms, namely, raw and recycled, as well as tubes and bottles. The results of this work show an increase in viscosity with an increase in the content of recycled PVC. Optimal tensile strength and impact strengths were determined using SEM fracture photo-micrography: hardness increases with the increasing density of the composite. In his article, Tatsuno [36] presented the results of plastics reinforced by carbon fibre. This study explained the behaviour of fibre deformation and press load during formation and the relationship between material temperature, press load and mechanical strength. The use of constant pressure on the press has been beneficial for obtaining good mechanical strength. Diaz and Ortega [37] presented a concept using rotary moulding, in order to reprocess polymers derived from cable waste. Hou, Ye and Mai [38] presented the results of the production of advanced composite elements for civil aircraft by the compression moulding method, using a polyetherimide composite reinforced with carbon fibre. As a result of pressed moulding, aileron ribs and hinges for civil aircraft with high load resistance were obtained. The use of both physical and chemical modifications improves the mixing effect of plastics and thus obtains the homogeneous properties of composite products [39]. Similar conclusions have been made in other works [40,41]. Another example of polymer plate production is the article by Vidales-Barriguete et al. [42]. The aim of their work was to analyse the physical and mechanical properties of mixed gypsum boards with waste plastic cable sheaths. The results of this work indicate a significant increase in the elasticity of plates with plastic waste when the reduction in cracking is taken into account, along with compliance with the minimum value of bending strength and a slight improvement in the heat conduction coefficient, resulting from lower energy demand. As stated in the study by Vidales-Barriguete, gypsum boards with the addition of plastic could be a suitable alternative to the gypsum boards on the market, contributing to sustainable construction, not only in new constructions but also in renovated buildings. Yet another article [42] presents an alternative material for acoustic barriers, anti-sound barriers, produced partly from plastic waste, rather than entirely from wood or steel. Materials made of recycled plastic do not differ significantly in utility, nor as regards conservation and aesthetic properties, but are also cheaper to manufacture and use (up to 194 USD/m^2^ vs. up to 269 USD/m^2^). A similar issue concerning the production of plates from secondary materials was discussed in the paper [43]. The boards made of wood and plywood were compared with the material obtained from recycled material from the paper industry, which also includes plastics. Each plate was obtained in the same way: the input material was placed in a mould, heated and pressed on a hydraulic press. The waste boards had similar mechanical properties to the agglomerated wood boards. However, they were distinguished by their higher flexibility and a significant difference in swelling. Water absorption tests showed that the waste board performed better than the agglomerated wood and plywood boards.

To the best of our knowledge, no research has focused on the development of secondary plastic plate manufacturing technologies that could be transferred into industrial manufacturing. In general, the above-mentioned research focussed on finding ways to improve certain properties of polymers by using fillers or modifiers in the injection and extrusion process as well as to apply sustainability and circular economy criteria to newly developed “eco-products”.

The aim of this work is to develop a solution for the recycling of cable waste by designing a procedure for converting it into solid sheets and examining its mechanical properties. The following tasks arise from the above purpose:Estimation of the fractions of individual plastics per batch of waste delivered;Manufacture of a plate-type product enabling the acquisition of research samples;Performing selected mechanical tests to assess the quality of the plate material obtained from recyclates along with an analysis and evaluation of the test results;A proposal for the use of recyclates from the dominant plastic fractions and fibrous composites, resulting mainly from cable waste.

## 2. Materials and Methods

Waste plastics, also known as PW, were supplied by the recycling company Eco Harpoon Sp. z. o.o with registered offices in Cząstków Mazowiecki (Poland). The share of waste plastics in the material delivered was determined on the basis of the organoleptic identification undertaken. The waste was in the form of cut and torn outer and inner wire covering. Figure 1 shows an example of the electric cable bundles provided by the company.

As mentioned above, the identification of plastics was carried out using a simplified method, the so-called organoleptic method which consists in observing the effects accompanying the combustion of a sample of the material in the burner flame. The combustion method was used, generally, to segregate the types of polymers, namely, thermoplastics, which, under the influence of temperature, soften, deform and melt, and duroplasts, which do not exhibit such characteristics. During the combustion tests, the following was established:(a)The flammability of plastic material;(b)The flame colour and colour scheme, i.e., the type of flame;(c)The behaviour of the material in the flame;(d)The smell of fumes after the sample is extinguished.

Due to the specific behaviour of PVC in the flame burner, the organoleptic method was used to quickly distinguish non-comminuted cable sheaths. The test resulted in separated (in Figure 1) PVC covers (right side) and only a few polysiloxane covers (left side). The organoleptic method is not an ideal method for identifying plastics, if only because of poor individual interpretation of tester; that is, the statements “irritating smell” and “sweet smell” can be quite vague when referring to test results and ambiguous for several independent testers. The organoleptic method was used only for the initial grouping of the material that may be present in the tested material. In order to check the content of the above-mentioned types of polymers, Fourier transform infrared (FTIR) spectroscopy was additionally performed using the Thermo scientific nicolet iS50 spectroscope, which is discussed in more detail in Section 3.1.

Subsequently, electric cables were subjected to pre-treatment processes, including shredding and separation processes, resulting in a fraction of the waste plastics (PW), namely, the so-called “speck”, being devoid of some metals, i.e., aluminium. Shredding stage was performed with the Wire Shredder Stokermill K750 (STOKKERMILL Recycling, Udine, Italy). Subsequently, speck was characterised in terms of its size and quantity using sieves with different mesh sizes ranging from 0.1 mm to 3 mm. Percentage distribution of individual fractions (together with the comminuted plastics) is shown in Figure 2b. Initial weight of speck was 463.101 g.

### 2.1. Sample Preparation 

As already mentioned in the previous section, so-called “speck” was used in the press moulding for the production of plates. If the “speck” comprises shredded polysiloxane insulations, it will remain in the same aggregate during plasticising and pressing, thus forming inclusions. Thick-walled plates were made using the press moulding technology. The following treatments can be distinguished in the technological process:-Preparation of the mould for the process, in particular securing all surfaces of the mould in contact with the processed material by using an anti-adhesive agent to protect it from sticking;-Preparation of the plastic for processing, i.e., heating in the heating chamber at a temperature of 60 °C for 2 h to evaporate moisture from the surface of the material;-Filling the mould with an appropriate amount of material in the form of the “regrind” obtained for testing and heating it to the appropriate temperature, as well as closing the mould with a punch plate with a preload;-Press moulding, such as the punch load with appropriate force for a specified time vis-à-vis the plasticised material;-Cooling the mould with the product, removing the punch plate, disassembling one side plate of the mould and then removing the product.

The technological station for the production of plates is shown in Figure 3. The test station consists of a metal mould with a section of resistance heaters embedded in it and an electronic heating system with temperature control, mounted on the table of a hydraulic press with a trigger acting on the punch plate. The task of the heating and temperature control system was to heat the mould to the temperature previously adopted by appropriate programming of the control system. The value of the mould temperature, as set and the current temperature of the processed material, measured in its geometric centre, was read on the front panel of the programmer. The total power of the heaters mounted in the form was 5.5 kW.

The parameters of the PVC sheet press moulding process were established, taking into account the following facts:-Glass transition temperature T_g_ is about 85 °C, for material with a high plasticiser content up to 75 °C;-In the temperature range of 140–170 °C, degradation of the plastic begins, and hydrogen is released.

The following parameters of process were adopted:-Material temperature T_w_ = 135 °C;-Heating time t = 1 h;-Pressure p = 10 MPa.

The temperature of the material before forming was measured in the geometric centre of the volume of backfilled-material-buried “speck”, while in the heating phase, the material was preloaded with the pressure coming from the weight of the punch, i.e., approximately 0.02 MPa. As a result of press moulding, a plate with a thickness of 1.4 cm, 17.5 cm wide and 20 cm long, as shown in Figure 4, was obtained. 

As a result of the pressing process, the plasticised material flowed with partial mixing of the particles of the plasticised material (Figure 4). However, one cannot talk about the homogenisation of the structure as it is in the plasticizing system of the extruder. During the heating, the fragmented particles of the cable sheaths were only adhesively connected with a small initial pressure. In the flow direction (Figure 5) of the material, the strength properties may be higher than in the transverse direction. In addition, the material is not homogeneous throughout the mass, so there may be local differences in the values of some properties. Waste covers probably come from different manufacturers and can be significantly different in terms of the amount and type of additives, including plasticizers.

### 2.2. Strength Tests of Samples from Waste Plastic 

#### 2.2.1. Test of Static Compressive Strength 

The static compression test was performed in accordance with the standard ISO 15527:2022 [44]. Samples with a square cross-section of 12 × 12 mm and a height of 5 mm were prepared for the compression tests. 

The yield point *R_e_* was calculated according to the following relationship:(1)$$R_{e}=\frac{F}{A}$$
where *F*—strength at yield point, and *A*—initial cross-sectional area of the sample.

#### 2.2.2. Test of Static Bend Strength

The static bend strength test was performed in accordance with the standard ISO 178:2019 [45]. For three-point bending tests, samples with a length of 120 mm and cross-sectional dimensions of 15 × 12 mm were prepared.

Bending strength *R_g_* was calculated according to the following formula:(2)$$R_{g}=\frac{M}{W}=3\cdot F\cdot l/2\cdot b\cdot h^{2}$$
where *M*—bending moment, *W*—strength index, *F*—maximum bending force, *l*—distance between supports, and *b, h*—width and height of the sample.

#### 2.2.3. Charpy Impact Test

The material impact tests were carried out on a Charpy apparatus. The Charpy impact test was performed in accordance with the standard ISO 179-1:2010 [46]. Samples of material with dimensions *l* × *b* × *h* = 120 × 15 × 10 mm were cut in the same way as the samples for bending tests. Data and results are given in instrument scaling units and then converted to the SI system. 

The impact strength was calculated according to the following relationship:(3)$$a_{n}=\frac{A_{n}}{b\cdot h}$$
where *b, h*—width and thickness of the sample, and *A_n_*—work needed to break the sample.

## 3. Results and Discussion

### 3.1. Identification of the Plastic 

The materials were identified and separated into fractions, and their masses were weighed (Table 1). In the batch delivered, plasticised polyvinyl chloride (PVC-P) predominated and constituted approximately 99.3%. The remaining part is polysiloxane (silicone) (SI) in the form of three bonded sheaths that insulate the electric wiring.

In order to accurately conduct the analysis of the tested waste cables, the FTIR analysis was performed. The absorption spectra ranged from 400 to 4000 cm^−1^ using an ATR detector with a resolution of 16 scans per spectrum and an optical resolution of 4 cm^−1^. In order to maintain the reliability and repeatability of the results, the ATR crystal was cleaned with alcohol-soaked tissue after each measurement. Before each measurement, the background spectrum was measured and collected. The tests were carried out in ambient conditions (room temperature 23 °C, air humidity 40%). The FTIR−ATR analysis was performed for six samples, and then the result was averaged. The measurement results are shown in Figure 6.

Describing from the left of the spectrum, the characteristic broadly distributed peak at 3380 cm^−1^ is characteristic of the stretching vibrations of the -OH group. The peaks at 2918 cm^−1^ and 1425 cm^−1^ are characteristic of the occurrence of the asymmetric stretching vibrations and deformation vibrations of the -CH_3_ group. The slightly lower peaks of 2851 cm^−1^ and 1259 cm^−1^ are characteristic of the occurrence of the -CH group, responsible for deformation and bending vibrations. The peaks at 2960 cm^−1^, 1071 cm^−1^ and in the range from 874 cm^−1^ to 796 cm^−1^ are responsible for the vibration of the -CH_2_ group. The present peak at 1719 cm^−1^ and 1018 cm^−1^ is characteristic of the stretching -CO group. The collected spectrum and its characterisation are consistent and confirm literature reports [47,48,49,50,51]. In summary, the FTIR−ATR analysis identified six polymer components and one calcium carbonate filler compound.

### 3.2. Compression Test Results 

On the basis of compression tests, yield point *R_e_* was calculated. The results of the measurements and calculations are presented in Table 2. The waveforms of compressive stresses are presented in Figure 7.

The standard deviation obtained from the test is S*_x_* = 0.5196, the critical value of the T-Student distribution, with k = n–1 degrees of freedom, is t*_α_* = 2.5706, and the significance level is α*_ist_* = 0.05.

The stress at yield point *R_e_*, in the compression test, on the samples taken from the plates, varies in the range from 14.1 to 15.6 MPa. The average value is *R_e_* = 15.0 ± 0.6 MPa. The measurement error is relatively small, less than 5%. Changes in the value of compressive stresses, as a function of deformation in a sample and used as an example, are shown in Figure 7. 

### 3.3. Bending Test Results

On the basis of bending tests, strength *R_m_* and yield point *R_e_* were calculated. The results of measurements and calculations are presented in Table 3.

The bending strength of the plate material varies from 0.8 to 1.16 MPa, while the value of the yield strength ranges from 0.40 to 0.56 MPa. The average values of the bending strength and yield point are as follows: *R_m_* = 0.94 ± 0.14 MPa, *R_e_* = 0.47 ± 0.07 MPa.

In the case of the values of *R_m_* as well as *R_e_*, measurement errors are relatively large, in the range of a dozen percent. The results of the load—displacement curves in three-point bending test are shown in Figure 8.

### 3.4. Impact Test Results

On the basis of tests carried out on the Charpy apparatus, the impact strength was calculated *a_n_*. The results of the measurements and calculations are shown in Table 4.

### 3.5. Static Tensile Test and Hardness

Results of the uniaxial stretching test are shown in Figure 9. Three stretching tests were performed on standard, paddle-shaped samples, according to the standard ISO 37:2005 [52]. Based on literature data [53], pure polyvinyl chloride reaches a value in the range of 0.785 to even 156 MPa, while Shore hardness ranges from 35 to 110MPa. It depends on the addition of plasticizers to the material. For example, Tygon polymer B-44-4X contains a small amount of phthalate plasticizers (orthophthalate), which reduce the intermolecular interaction and increase the mobility of polymer chains. Thus, they reduce hardness and increase its flexibility. Comparing the results of the static tensile test with other polymers, such as Silopren E VPI 4036 G, Tygon polymer R-40, Siloprene E3078, etc., the properties of the obtained material are similar to the commercial Tygon Polymer R-1000 [53]. Its tensile strength is about 8 MPa, and the hardness is 40 Shore. The presented graphs show that the tensile strains are elastic to the failure point with deformations exceeding 100% (Table 5). Tensile strength is typical for this group of materials. 

The measurement of the hardness of elastomeric materials such as various types of rubbers and plastics can be performed using the Shore method, in accordance with ISO 7619-1:210 [54]. The Sauter Hardness Test HDD 100-1 was used for hardness testing. The Shore hardness (D scale) was 36.

## 4. Conclusions

On the basis of the test results obtained for recycled PVC, it was found that the products manufactured from them after the shredding process should have the structure of solid or perforated plates and thus will be resistant to compressive stresses. Yield strength *R_e_* for PVC varies in a very wide range from 0.3 Pa to approximately 70 MPa depending on the content of plasticisers which can be compared to the results obtained in [55,56]. The strength value of 15.0 MPa, obtained in the tests, can be considered quite good, suitable for materials of various grades and probably of various manufacturers, which were additionally used for several years or more. During this time, PVC insulations were exposed to overheating and environmental influences, such as UV radiation.

A significant influence of plasticisers on mechanical properties was observed. For example, the shear modulus of elasticity (Young) of PVC at T = 20 °C was more than 1200 MPa for 0% DOP content (di-octyl phthalate) and approximately 45 MPA for the 30% DOP content, whereas for a 50% DOP content, only 15–20 MPa, similar to [57]. There are also PVC grades on the market with E = 3000 MPa. 

Similar to the results of the compression tests, the result *R*, obtained in the bending test *S_m_* = 0.94 MPa, is typical for plasticised products, as described in [58]. For non-plasticised products, *R_m_* can be about 90 MPa [59]. The Charpy impact strength of the product (plate) of a_n_ = 5.1 kJ/m^2^ is comparable to materials made of granulate as described in [60,61,62].

The results obtained from the analysis performed by FTIR−ATRinfrared spectroscopy and analysed with the available library database confirmed the presence of polymers such as polyvinyl chloride (PVC), silicone, poly(ethylene terephthalate) (PET), polypropylene (PP), polyethylene (PE), nylon and trace amounts of calcium carbonate, which is a fairly common inorganic filler in cables. The remaining unidentified noise collected in the absorption spectrum is most likely due to impurities or the presence of other polymers appearing in trace amounts.

The test results confirm the possibility of producing simple structures such as a solid plate, despite the fact that the material is neither perfectly plasticised nor homogeneous. Under manufacturing conditions, it is possible to plasticise the material in the extrusion process, and, at a later stage, the material stream with a slightly lower temperature can be widened with sections of rollers or pressed without additional heating. It was also stated that in the case of press moulding under laboratory conditions, PVC cannot be heated to a sufficiently high temperature for a long time due to the possibility of it degrading. The obtained results encourage research aimed at developing a technology for making prototype products. Despite the worse strength and processing parameters of recyclates, through skilful dosing of reinforcing materials, composites with the required physical and chemical properties can be obtained.

Due to the fact that the mechanical properties are not very high, compared to other materials, the use of the obtained plates is, of course, limited, and they should only be used in the case of low bearing. However, taking into account other physico-chemical properties, such as low electrical conductivity or low wettability, as well as the final shape (plates or blocks), the authors suggest that such objects could be used as electrical insulators (which, in fact, this material originally was) or as water shields such as windscreen wipers, roofing, etc.).

Based on the research results presented, the following conclusions can be drawn:-Using the method of plastic consolidation, it is possible to obtain solid plates or with recesses from waste materials, meeting the mechanical requirements of some types of products, e.g., platforms;-Materials after plastic consolidation show increased plasticity compared to the material of plates made of the original granulate;-The microstructural tests carried out showed no defects inside the analysed plates, such as cracks or delamination, which confirms the correct selection of pressing parameters.

It should be noted that the plates used in water and sewage infrastructure, road infrastructure, etc., under real conditions, are mainly loaded with compressive and bending forces. The results obtained from the research are promising for the applications mentioned.

## Figures and Tables

**Figure 1 materials-15-09019-f001:**
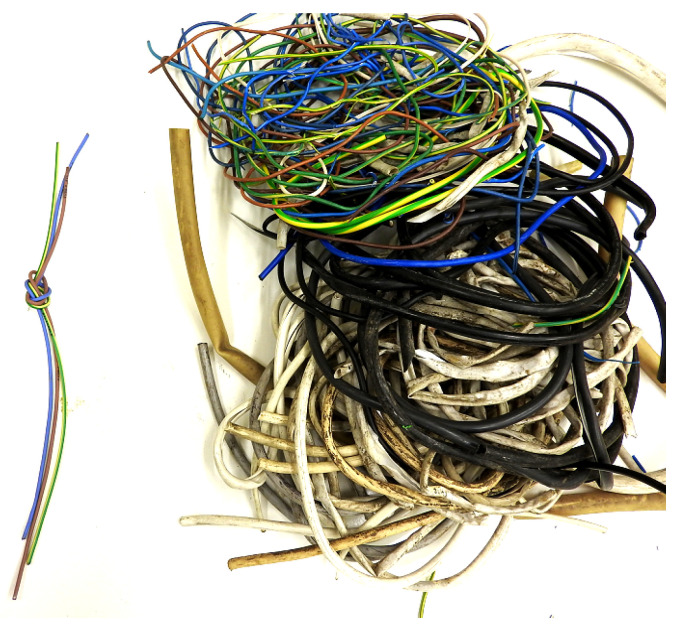
Examples of scrap electronic equipment delivered for testing.

**Figure 2 materials-15-09019-f002:**
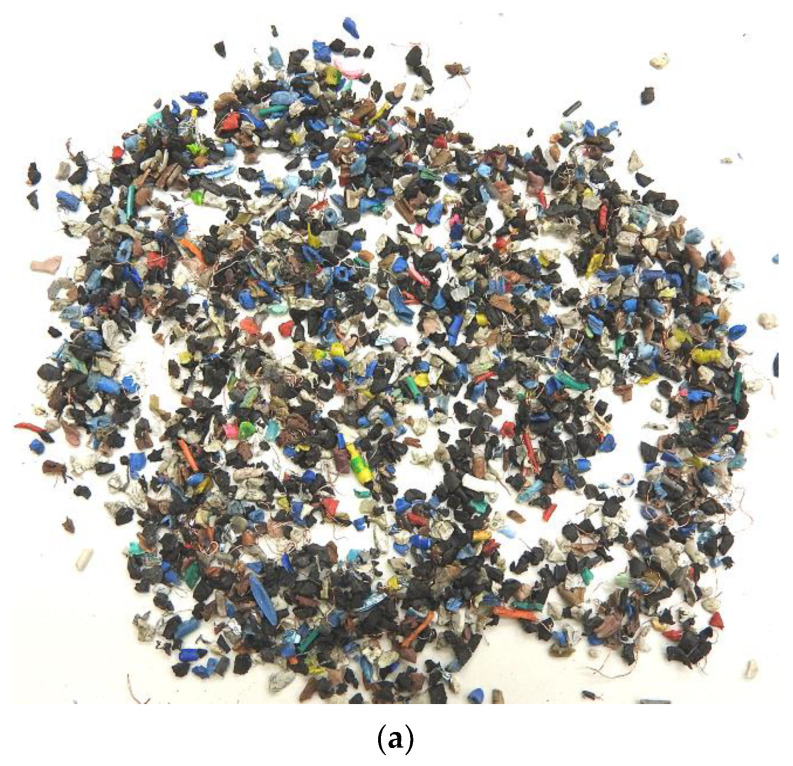
(**a**) Input material (“speck”) to the moulding process, obtained in the shredding process. (**b**) Percentage distribution size of the fractions by particle size.

**Figure 3 materials-15-09019-f003:**
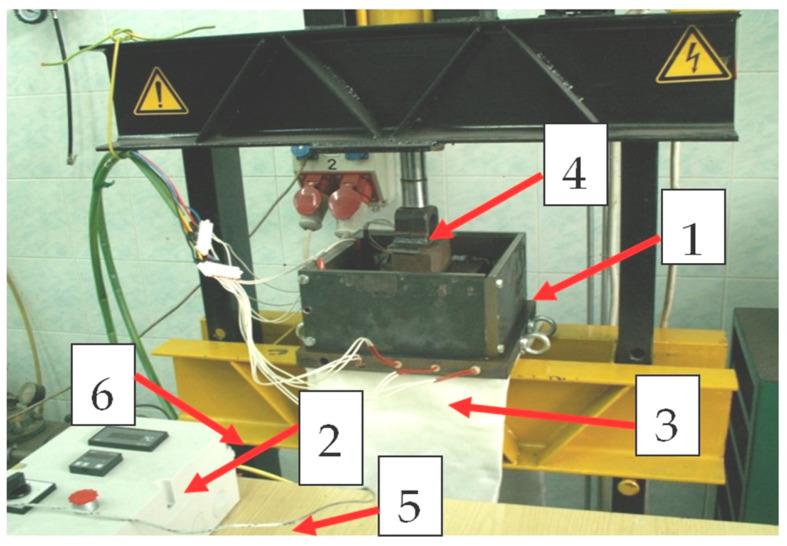
Test station for the production of plates obtained from the recycling of cable waste: 1—metal mould, 2—heating system with temperature control, 3—resistance heaters, 4—pressure pin of the hydraulic cylinder, 5—control thermo-couple cable, 6—base of the hydraulic press.

**Figure 4 materials-15-09019-f004:**
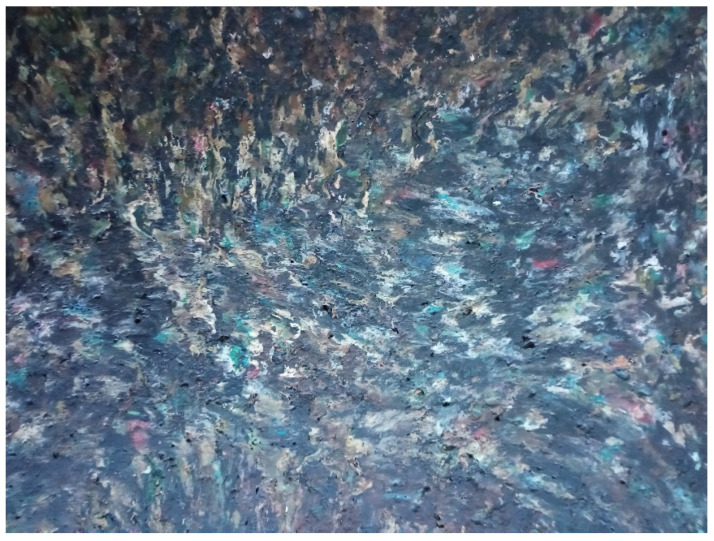
Plate obtained in the press moulding process.

**Figure 5 materials-15-09019-f005:**
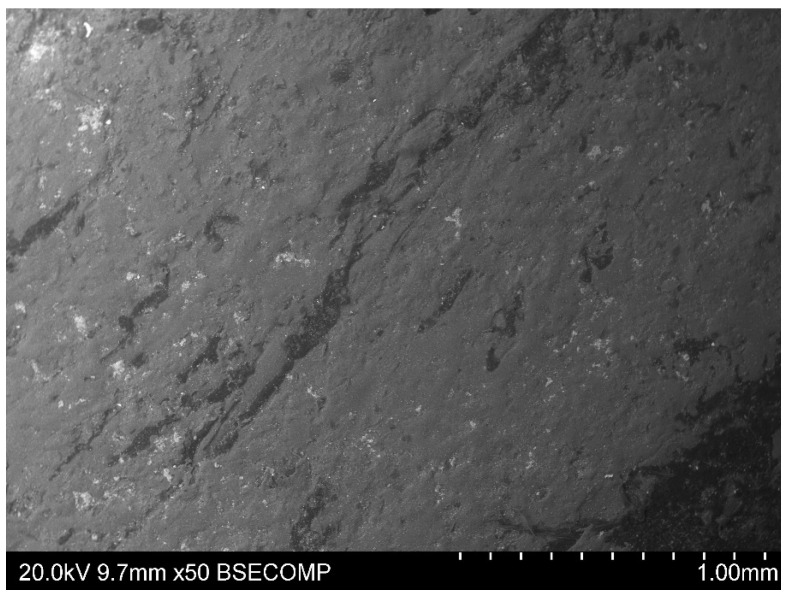
Scanning electron microscope (SEM) micrograph of the plate obtained in the press moulding process.

**Figure 6 materials-15-09019-f006:**
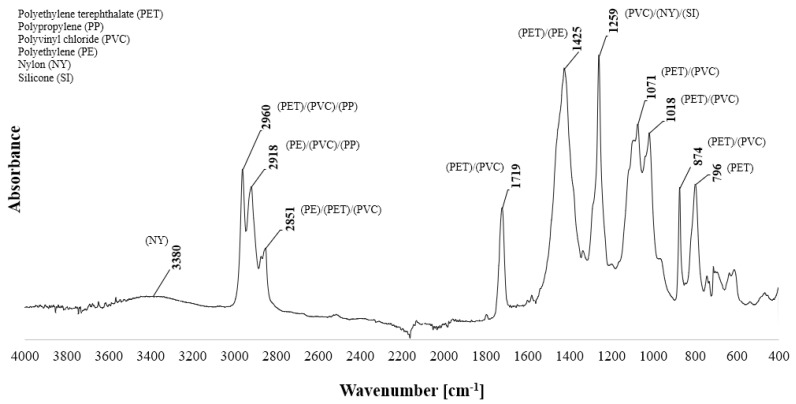
FTIR−ATR spectrum for recycled cable waste.

**Figure 7 materials-15-09019-f007:**
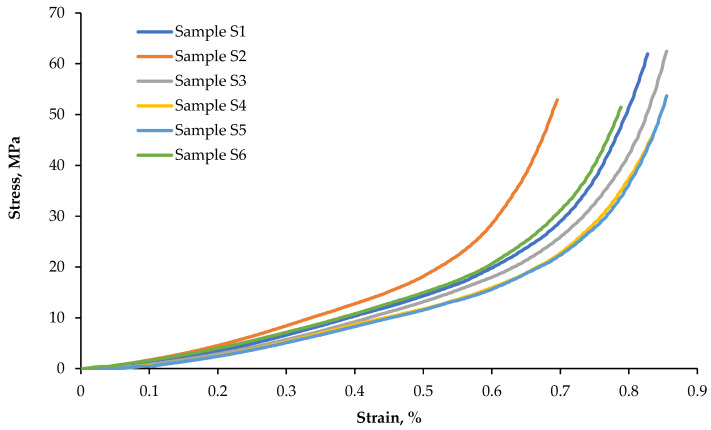
Waveforms of compressive stresses as a function of the deformation of specimens S1–S6.

**Figure 8 materials-15-09019-f008:**
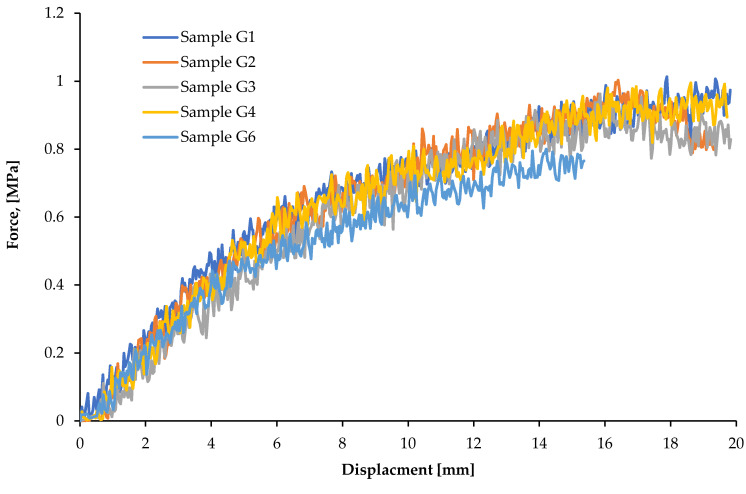
Load—displacement curves in three-point bending test (samples G1—G6).

**Figure 9 materials-15-09019-f009:**
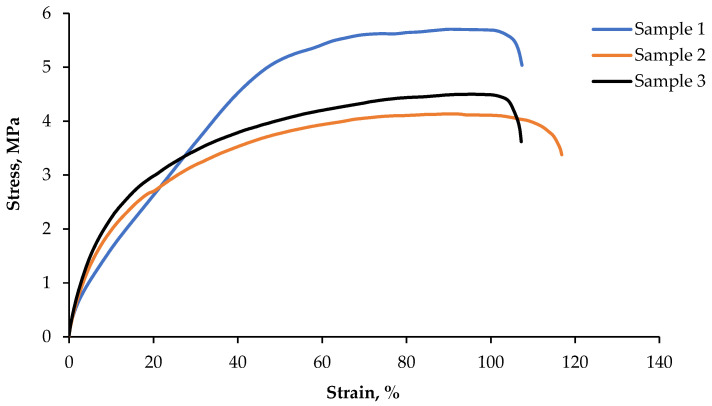
Tensile curves of samples made from the test plate.

**Table 1 materials-15-09019-t001:** Results of identification and mass measurements of the waste delivered for testing.

Item			Share		
	Effects—Burning in the Burner Flame	Identification	Quantity pcs	Weight, g	Percent, %
1	The density is higher than that of water; self-extinguishing; does not drip; yellow centre of the flame with a green border at the bottom; the smell of hydrochloric acid; when removed from the flame—white smoke	Polyvinyl chloride (PVC)	126	847.6	99.3
2	The density is higher than that of water; burns with a bright yellow flame; no smell; after extinguishing—white ash	Polysiloxane (silicone) (SI)	3	6.0	0.7
In total			129	853.6	100

**Table 2 materials-15-09019-t002:** The results of the calculation of the plate yield point.

Item	Compression			
	Sample No.	F*_pl_* (kN)	ΔL*_pl_* (mm)	R*_e_* (MPa)
1.	S1	2.96	6.48	15.0
2.	S2	2.46	6.52	14.1
3.	S3	2.38	6.42	15.1
4.	S4	2.41	6.47	15.6
5.	S5	2.24	6.58	15.4
6.	S6	2.22	6.55	14.9

Final result: R*_e_* = 15.0 ± 0.6 MPa.

**Table 3 materials-15-09019-t003:** Results of calculations of the bending strength and the yield point of plates.

Item	Three-Point Bending Test						
	Sample No.	F*_pl_* (kN)	ΔL*_pl_* (mm)	F*_max_* (kN)	ΔL*_max_* (mm)	R*_m_* (MPa)	R*_e_* (MPa)
1.	G1	6.43	4.01	13.67	17.01	0.91	0.40
2.	G2	7.43	3.72	15.08	16.02	0.96	0.41
3.	G3	6.33	5.01	12.52	16.23	0.93	0.46
4.	G4	7.79	5.12	15.80	16.84	0.90	0.52
5.	G5	8.47	4.89	16.73	16.88	1.16	0.56
6.	G6	7.89	4.82	13.65	17.02	0.80	0.48

Final result: R*_m_* = 0.94 ± 0.14 MPa for: S_x_ = 0.1367, t*_α_* = 2.5706. Final result: R*_e_* = 0.47 ± 0.07 MPa for: S_x_ = 0.0621, t*_α_* = 2.5706.

**Table 4 materials-15-09019-t004:** Results of measurements and calculations of impact strength according to Charpy.

Item	Charpy’s Impact Test				
	Sample No.	A*_n_* (J)	*b*_x_*h* (cm^2^)	a*_n_* (J/cm^2^)	a*_n_* (kJ/m^2^)
1.	U1	0.78	1.62	0.481	4.8
2.	U2	0.75	1.53	0.490	4.9
3.	U3	0.90	1.60	0.562	5.6
	U4	0.75	1.53	0.490	4.9
5.	U5	0.80	1.63	0.491	4.9
6.	U6	0.85	1.53	0.556	5.6

Final result: a_n_ = 5.1 ± 0.4 kJ/m^2^ for: S_x_ = 0.3768, t_α_ = 2.5706.

**Table 5 materials-15-09019-t005:** Mechanical properties, established on the basis of the tensile test.

Item	R*_m_* (MPa)	A (%)
Sample 1	5.70	101
Sample 2	4.14	110
Sample 3	3.47	102
Average	4.44	105

## Data Availability

Data sharing is not applicable to this article.

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
