# Peer review of "Determining the Mechanical Properties of Solid Plates Obtained from the Recycling of Cable Waste"

_materials, 2022, doi:10.3390/ma15249019_

Round 1

Reviewer 1 Report

1. The introduction part is too repeated, should be condensed key points, highlight the innovation of the article.

2. As we all know, waste cable sheath plastics include polyvinyl chloride (PVC), polyethylene (PE), polyperfluorinated ethylene propylene (F46), nylon, polyolefin and so on. The author should make an in-depth structural analysis of the collected waste cable sheaths. In addition, the waste cable sheaths contain a small amount of inorganic filler, such as calcium carbonate, etc. Do these components have any effect on their mechanical properties?

3. As can be seen from Figure 4, the waste cable sheath plastics particles only melted and fused under the action of hot pressing. The internal morphology and structure should be observed by SEM. In addition, if the waste cable sheath plastic is melted and then hot pressed, will the sample have better mechanical properties?

4. The standard against which the charpy impact and static bend strength of samples are tested shall be given.

5. The identification results and methods listed in Table 1 are unscientific, and the authors should use more analytical means such as FT-IR, NMR or MS to support these results.

6. Waveforms of compressive stresses and tensile curves for other samples should also be given in Figure 5 and Figure 6.

7. The conclusion section is poorly organized, and the author should rerefine the relevant important results and shortcomings of the study, and point the way for future research in this area.

8. The author should cite more references from the journal of Materials.

9. The grammar, spelling and formatting of the text should be completely revised.

10. The format of references should be uniform.

Author Response

Answers are in the attachment

Reviewer 2 Report

The introduction is too long and uninteresting because it talks about obvious things about polymers.

All part of the introduction related to the paper topics should be located in its own section.

The organoleptic characterization, although it is used in laboratory tests, is not optimal for a scientific study. Perhaps FTIR characterization would be better.

The preparation of the samples is too descriptive, missing important details such as the type and shape of the test sample. What are the dimensions of each specimen for each type of test?

Why a time of 1 hour in the oven with a thickness of only 1.4 cm?. I propose to use ASTM or ISO standards, instead of the specific standards of each country (in this case Poland). Formulas 1 and 2 are repeated.

I doubt that the composition of the sample can be obtained, with any significant precision, by means of organoleptic techniques.

The statistical study of the compressive strength is not enough clear.

Figure 5.- deformation units are missing.

The section on static tensile and hardness is too extensive and focuses very little on the discussion of the results obtained.

I believe that the paper is not appropriate to be published in Materials, there is not thermal characterization techniques, SEC, etc... to compare the results with results obtained by samples made with fresh polymers

Author Response

Answers are in the attachment

Round 2

Reviewer 1 Report

The authors have made sufficient improvements to the manuscript (materials-2056811 ) according to the revision comments, and therefore it is recommended that it be published in Materials.

Author Response

Dear Reviewer,

            Thank you very much on behalf of myself and the other co-authors of the article  for taking the time, reviewing the article and pointing out a point for improvement.

Reviewer 2 Report

The article has improved significantly with the modifications proposed by the authors. Despite all this, I think that the description of the FTIR is not accurate enough, it only describes bands but does not assign them to any of the components of the samples. Therefore, I think the authors should improve this description and assign each band to the different components

Author Response

Dear Reviewer,

The answer to the comment is in the attachment
